# Symptomatic and asymptomatic enteric protozoan parasitic infection and their association with subsequent growth parameters in under five children in South Asia and sub-Saharan Africa

**Rina Das**[1,2]*, **Parag Palit**[1,3], **Md. Ahshanul Haque**[1], **Myron M. Levine**[4], **Karen L. Kotloff**[4], **Dilruba Nasrin**[4], **M. Jahangir Hossain**[5], **Dipika Sur**[6], **Tahmeed Ahmed**[1,7,8], **Robert F. Breiman**[9◉], **Matthew C. Freeman**[2◉]*, **A. S. G. Faruque**[1◉]

1 Nutrition Research Division, International Centre for Diarrheal Disease Research, Bangladesh (icddr,b), Dhaka, Bangladesh, 2 Gangarosa Department of Environmental Health, Rollins School of Public Health, Emory University, Atlanta, Georgia, United States of America, 3 University of Virginia School of Medicine, Charlottesville, Virginia, United States of America, 4 University of Maryland School of Medicine, Baltimore, Maryland, United States of America, 5 Medical Research Council Unit, London School of Hygiene & Tropical Medicine, Fajara, The Gambia, 6 National Institute of Cholera and Enteric Diseases, Kolkata, West Bengal, India, 7 James P. Grant School of Public Health, BRAC University, Dhaka, Bangladesh, 8 Department of Global Health, University of Washington, Seattle, Washington, United States of America, 9 Hubert Department of Global Health, Rollins School of Public Health, Emory University, Atlanta, Georgia, United States of America

◉ These authors contributed equally to this work.
* rina.das@emory.edu, rina.das@icddrb.org (RD); matthew.freeman@emory.edu (MCF)

## Abstract

### Background

*Entamoeba histolytica*, *Giardia*, and *Cryptosporidium* are common intestinal protozoan parasites that contribute to a high burden of childhood morbidity and mortality. Our study quantified the association between intestinal protozoan parasites and child anthropometric outcomes among children under-5.

### Methods

We analyzed data from 7,800 children enrolled in the Global Enteric Multicenter Study (GEMS) across seven study sites that were positive for intestinal protozoan parasites between December 2007 and March 2011. Parasites were assessed using stool immunoassays (ELISA). We applied multiple linear regression to test the association between any or concurrent parasite and child anthropometric outcomes: length/height-for-age (HAZ), weight-for-age (WAZ), and weight-for-length/height (WHZ) z-score after 60 days of enrollment. Models were stratified by diarrheal symptoms, driven by the study design, and adjusted for potential covariates.

**Data Availability Statement:** A publicly available GEMS dataset was analyzed in this study. This data

can be obtained here: ClinEpiDB (https://clinepidb.org/ce/app/workspace/analyses/DS_841a9f5259/new/variables/PCO_0000024/ENVO_00000009). Following the thorough review and approval process by the ClinEpiDB study team, we have obtained official data access from ClinEpiDB, the responsible entity for managing the GEMS data repository.

**Funding:** This work was supported, in whole or in part, by the Bill and Melinda Gates Foundation (grant number: INV-002050) to ASGF. Under the grant conditions of the Foundation, a Creative Commons Attribution 4.0 Generic License has already been assigned to the Author's Accepted Manuscript version that might arise from this submission. The funders had no role in study design, data collection and analysis, decision to publish, or preparation of the manuscript.

**Competing interests:** The authors have declared that no competing interests exist.

## Findings

During the follow-up at day 60 after enrollment, child anthropometric outcomes, among the asymptomatic children showed, negative associations between *Giardia* with HAZ [β: -0.13; 95% CI: -0.17, -0.09; p<0.001] and WAZ [β -0.07; 95% CI: -0.11, -0.04; p<0.001], but not WHZ [β: -0.02; 95% CI:-0.06, 0.02; p = 0.36]; *Cryptosporidium* with WAZ [β: -0.15; 95% CI: -0.22, -0.09; p<0.001] and WHZ [β: -0.18; 95%CI: -0.25, -0.12; p<0.001], but not with HAZ [β: -0.03; 95% CI: -0.09, 0.04; p = 0.40]. For symptomatic children, no associations were found between *Giardia* and anthropometry; negative associations were found between *Cryptosporidium* with HAZ [β: -0.17; 95% CI: -0.23, -0.11; p<0.001], WAZ [β: -0.25; 95% CI: -0.31, -0.19; p<0.001] and WHZ [β: -0.23; 95% CI: -0.30, -0.17; p<0.001]. Among the asymptomatic 24–59 months children, *Giardia* had a negative association with HAZ [β: -0.09; 95% CI: -0.15, -0.04; p = 0.001]. No significant associations were found between *E. histolytica* with child growth.

## Conclusions

While some studies have found that *Giardia* is not associated with (or protective against) acute diarrhea, our findings suggest that it is associated with growth shortfall. This observation underscores the need for preventive strategies targeting enteric protozoan parasites among young children, to reduce the burden of childhood malnutrition.

## Author summary

Intestinal protozoan parasites such as *Entamoeba histolytica, Giardia*, and *Cryptosporidium* are significant causes of childhood morbidity and mortality. A study analyzed data from 7,800 children enrolled in the Global Enteric Multicenter Study (GEMS) across seven study sites who tested positive for these parasites using stool immunoassays. The study aimed to quantify the association between intestinal protozoan parasites and child anthropometric outcomes among children under the age of 5.

The results of the study showed negative associations between *Giardia* and weight-for-age and length/height-for-age among asymptomatic children. Similarly, *Cryptosporidium* was negatively associated with weight-for-age and weight-for-length/height among asymptomatic children and with weight-for-age, weight-for-length/height, and length/height-for-age among symptomatic children. No significant associations were found between *Entamoeba histolytica* and child growth.

The study findings suggest that while *Giardia* may not cause acute diarrhea, it is associated with growth shortfall among young children. These observations highlight the need for preventive strategies to target enteric protozoan parasites among young children to reduce the burden of childhood malnutrition.

## Introduction

Intestinal infections are the leading cause of childhood morbidity and mortality in children under 5 [1], caused by numerous microorganisms. Protozoa, particularly *Entamoeba*, *Giardia*, and *Cryptosporidium* [2], infecting 450 million individuals annually [3]. The infection leads to

diarrhea, abdominal pain, vomiting, weight loss, and the long-term sequelae of childhood growth faltering [4]. Malnutrition is known to impair cellular immunity, which is an important risk factor for infection by these enteric protozoan parasites [5]. *Entamoeba histolytica* is associated with moderate-to-severe diarrhea (MSD) and increased mortality among children in African countries and has been reported to exert adverse effects on child growth and development [6]. Infections with *Giardia* are quite frequent and are one of the most common causes of diarrhea, but generally occur without clinical symptoms, though it can affect nutritional status due to diarrhea, loss of appetite, reduced absorption of proteins, vitamins A, B12, and lactose [7]. Children in resource-poor settings are particularly at risk of infections with *Cryptosporidium*, which has been reported to be the second leading cause of diarrhea-related mortality among under-5 in LMICs [8].

Though intestinal parasitic infection is widespread and often asymptomatic for diarrhea, their impact on growth in early childhood is largely uncharacterized. In the Global Enteric Multicenter Study (GEMS) across seven global study sites, the presence of *Cryptosporidium* was significantly associated with a greater decline in linear growth than in those without *Cryptosporidium* [9], but no analysis has been conducted on asymptomatic children who were positive for intestinal protozoan parasites. In the MAL-ED birth cohort study, asymptomatic infections by *Giardia* were found to be associated with growth faltering in the first 2 years of life [10], but no association was observed among the older children with parasitic infection and child growth. The mechanism by which parasitic disease impairs child growth is not fully understood but is thought to be related to host systemic responses to infection, disruption of host intestinal absorptive processes, and anemia [11]. Intestinal protozoan parasitic infections, disproportionately affect children; however, little is known about the impact of parasitic disease on growth in older children. Official figures may only represent a fraction of the true incidence of symptomatic cases and are often underdiagnosed and under-reported [12]. In addition, the proportion of asymptomatic carriers and subclinical infections is unknown due to the limited sensitivity of conventional (e.g., microscopy) diagnostic tests and the lack of large community surveys [12].

Here we assessed the impact of both symptomatic and asymptomatic enteric protozoan parasite infections on childhood growth among under-5 children in the GEMS dataset. Little is known about whether child growth faltering is enteric protozoan specific, and determining whether child malnutrition varies for different enteric protozoan parasites has implications for the implementation and evaluation of programs designed to improve child health.

## Methods

### Ethics statement

Before implementing the GEMS, the clinical procedure, consent forms, CRFs (case report forms), field procedures, and other supporting materials were approved by the local site-specific ethics committees and the ethics committee of the University of Maryland School of Medicine. The committees and their collaborating partners from other institutions oversee each site who had their IRB approvals. The collaborating institutes of 7 sites were, International Center for Diarrheal Disease Research, Bangladesh (icddr,b) in Bangladesh, National Institute of Cholera and Enteric Diseases (NICED) in India, Aga Khan University in Pakistan, Medical Research Council Unit, The Gambia in Gambia, CDC/Kenya Medical Research Institute (KEMRI) Research Station in Kenya, Centre pour le Development des Vaccines du Mali (CVD-Mali) in Mali, Centro de Investigação em Saúde de Manhiça (CISM) in Mozambique. The signed informed consent forms for the children's inclusion in the study were collected from the children's parents/guardians (both sick cases and healthy controls).

## Study design

GEMS was a prospective, age-stratified, matched case-control study conducted from December 2007 to February 2011 in seven study sites across sub-Saharan Africa (The Gambia, Mali, Mozambique, and Kenya) and South Asia (Bangladesh, India, and Pakistan). [13] Under-5 children of the Demographic Surveillance System (DSS) catchment area, presenting to the Sentinel Health Center with MSD admitted within 7 days of acute illness onset were considered cases. Age, sex, and community-matched healthy children without diarrhea for the previous 7 days randomly selected from the DSS were enrolled as controls. Nutritional assessments based on weight, length/height, and mid-upper arm circumference (MUAC) were performed at the time of enrollment [13].

Approximately 60 days after enrollment, GEMS field workers visited the household of each enrolled child (acceptable range, 50–90 days) [13]. We used the anthropometry data and details of in-between comorbidity data (malaria, typhoid, pneumonia, diarrhea, and dysentery) from these follow-up household visits for our analysis.

## Specimen collection and laboratory procedure

The GEMS protocol incorporated traditional bacterial culture, the growth of the pathogens, and characterizes them further for virulence, and serologic features, as described elsewhere [14]. *Giardia*, *E. histolytica*, and *Cryptosporidium* were detected using immunoassays (ELISA) available commercially from TechLab, Inc and according to the manufacturer's protocols [14].

## Outcome variable

The primary measure of growth in our analyses was the height/length-for-age z-score (HAZ), weight-for-age z-score (WAZ), and weight-for-height z-score (WHZ). In our study, we used baseline (after rehydration in case of MSD) and endline HAZ, WAZ, and WHZ from enrolment to follow-up for the *E. histolytica*, *Giardia*, and *Cryptosporidium*-positive children enrolled in GEMS (Fig 1). As the comparison group, we used the baseline and endline follow-up anthropometry data for the *E. histolytica*, *Giardia*, and *Cryptosporidium* negative children.

## Variables of interest

**Anthropometry measurements.** Height and weight were measured at enrollment and the 60-day follow-up visit for each child, and details of measuring methods were described

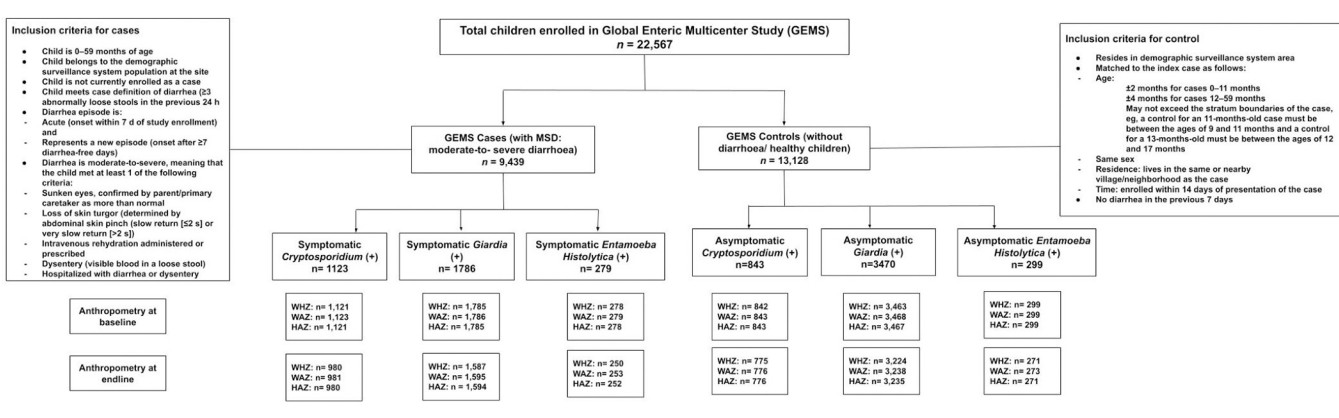

**Fig 1. Study flow diagram.**

elsewhere [8]. Using the WHO Child Growth Standards as the reference population, the HAZ/ LAZ, WAZ, and WHZ were measured using a WHO SAS macro [13].

**Moderate-to-severe Diarrhea (MSD).** MSD was defined as new and acute diarrhea ($\geq 3$ abnormally loose stools within the past 24 hours that started within the previous 7 days following at least 7 diarrhea-free days, with at least one of the following criteria for MSD: dehydration based on the study clinician's assessment (sunken eyes; decreased skin too, or; or intravenous rehydration administered or prescribed); dysentery (visible blood in stools reported by the mother or observed by the study team); or hospitalization with diarrhea or dysentery [13].

**Vomiting, fever, and dysentery.** Vomiting 3 or more times per day, and fever (at least 38˚C or parental perception) determined by retrospective and dysentery (visible blood in stools) assessed by the attending clinician [13].

**Asymptomatic children.** Age, sex, and community-matched healthy children without diarrhea for the previous 7 days randomly selected from the same DSS were enrolled as asymptomatic children [8].

**Breastfeeding.** Breastfed referred to both exclusive and partially breastfeed children under 2.

**Sociodemographic information.** Included data about the child's household including the mother as a primary caretaker, primary caretaker's education (illiterate/ literate), household size (number of children <5 years of age) were considered explanatory variables. Variables addressed WASH (before nursing or preparing baby food; after cleaning a child who defecated), the main source of drinking water (tube well water/ non-tube well water), sanitation facilities (toilet facility for disposal of human fecal waste available/ no facility), and the use of handwashing materials (water with soap/ without soap). Households were categorized based on the wealth quintiles as socioeconomic status (SES) (poor, lower-middle, middle, upper-middle, and rich) [15] by using principal component analysis. The survey includes data on asset indicators that can be grouped into three types: household ownership of consumer durables(clock/watch, bicycle, radio, television, bicycle, sewing machine, refrigerator, car); characteristics of the household's dwelling(about toilet facilities, the source of drinking water, rooms in the dwelling, building materials used, and the main source of lighting and cooking); and household land ownership [15]. Individuals were sorted by the asset index and established cut-off values for percentiles of the population. Then the households were assigned to a group based on their value on the index. For expository convenience, they refer to the bottom 20% as "poorest," the next 20% as "lower middle," the next 20% as "middle", the next 20% as "upper middle" and the top 20% as "richest," but this classification does not follow any of the usual definitions of poverty [15].

## Statistical analysis

We reported the child and household-level characteristics by using mean and standard deviation for continuous variables and frequency as a percentage for categorical variables to summarize the data. Chi-square and proportion tests were used to see the association between two categorical variables and the t-test was used to see the mean difference between two groups for symmetric distribution. To assess the association between the intestinal protozoan parasites at baseline and the change in the child's HAZ, WAZ, and WHZ in the subsequent ~60 days, we used a generalized linear model, where the explanatory variable was the presence of intestinal protozoan parasites (*Cryptosporidium*, *Giardia* and *E. histolytica*) and the outcome variable was (HAZ, WAZ, and WHZ). All the factors include age, gender, breastfeeding status, primary caretaker's education, number of under-5 children at the house, WASH, co-pathogens (ETEC,

EAEC, *Shigella*, *Campylobacter*, and Rotavirus), comorbidity, time (since it was a repeated measured data, we adjusted the variable [time: 0 and 1] as co-variate; the anthropometry was taken in two-time points: on enrollment = 0 and day ~60 follow up = 1), and study site, suggesting the association with the outcome as indicated in the literature were chosen for multi-variable modeling. Separate models were constructed to assess the association of each enteric protozoan parasitic infection with a child's HAZ, WAZ, and WHZ for symptomatic and asymptomatic infections. A similar model was performed for the detection of the association of protozoan parasite co-pathogens and child growth parameters after adjusting for the potential covariates. We checked for potential effect modifiers in the final model by checking the interactions. Initially, we checked the Mantel-Haenszel test to detect the heterogeneity of the association by different pathogens. A likelihood-ratio test (LRT) was performed and found that without the interaction term, the model had a better fit. The variance inflation factor (VIF) was calculated to detect multicollinearity, and no variable with a VIF value greater than 5 was identified in the final model. We estimated the β coefficient and its 95% CI to describe the precision of the point estimate. A p-value of $< 0.05$ was considered statistically significant and STATA 17.0 IC (Stata Corp LLC, College Station, TX) was used to analyze the data.

## Results

Among children with MSD, the detection of *Cryptosporidium* spp. was the highest in Mozambique (16.8%) and Indian (16.7%) study sites. In Mali, *Giardia* was found in 27.9% of children (Fig 2). Among the children without symptoms, the detection of *Cryptosporidium* spp. was the highest in India (10.7%) and the lowest in Bangladesh (4.0%). In Mali, *Giardia* was found in 37.5% of asymptomatic children. *Giardia* was detected in a higher proportion of asymptomatic children (healthy children enrolled as control from the community) in all sites than symptomatic MSD children (Fig 3).

Baseline anthropometric measurements (WAZ mean ±SD: -1.08± 1.32 vs -1.51± 1.39 and WHZ: -0.47± 1.49 vs -1.06± 1.49) showed a statistical difference (p-value <0.001) between

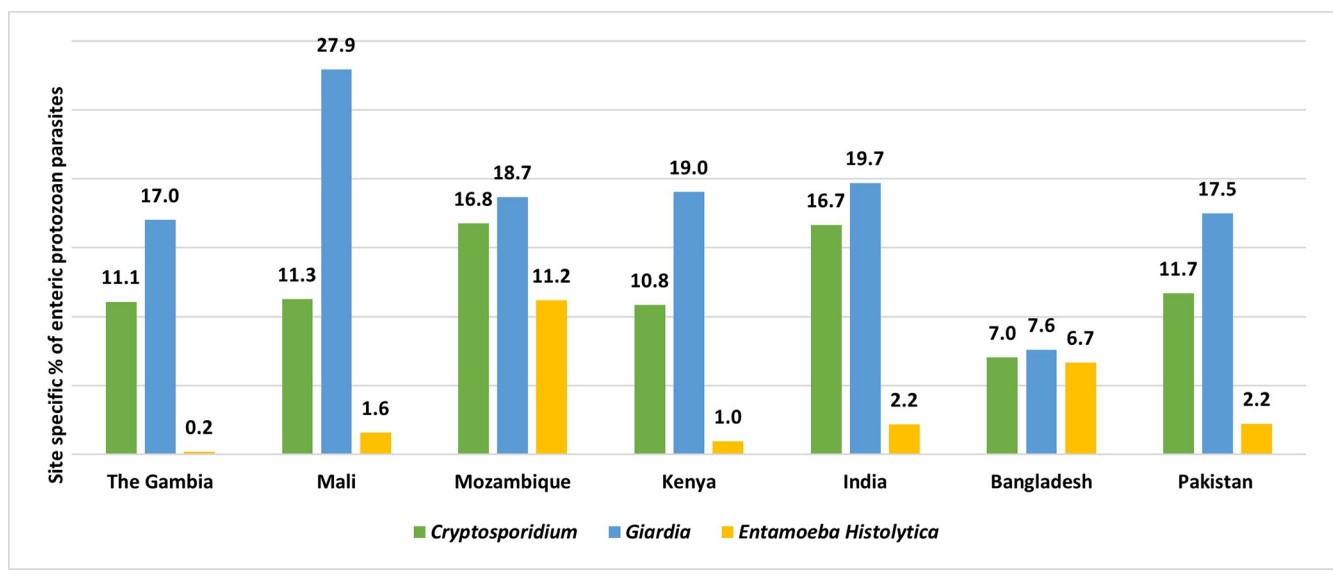

**Fig 2. Site-Specific percentage of enteric protozoan parasites (*Cryptosporidium*, *Giardia*, and *Entamoeba histolytica*) isolated from the stool of under 5 children among the symptomatic MSD cases.**

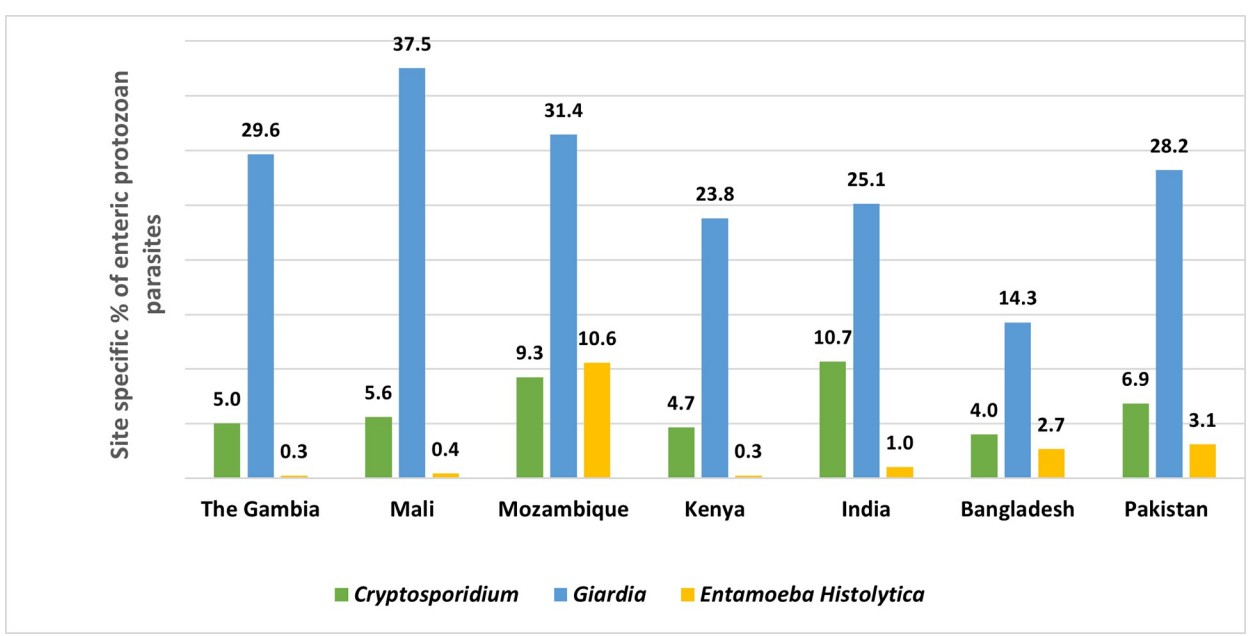

**Fig 3. Site-Specific percentage of enteric protozoan parasites (Cryptosporidium, Giardia, and Entamoeba histolytica) isolated from the stool of asymptomatic under 5 children (healthy children enrolled as control from the community).**

asymptomatic and symptomatic children respectively (Table 1). Symptomatic MSD children were more often (82.3%) using non-tube well water as the main source of drinking water compared to asymptomatic healthy children (82.3% vs 77.2%, p-value <0.001). Non-sanitary latrines were available for 6.4% of households. *Cryptosporidium* (11.9% vs 6.4%), *E. histolytica* (2.9% vs 2.3%), ETEC (11.3% vs 7.4%), rotavirus (18.5% vs 3.9%), and *Shigella* (11.8% vs 1.8%) were more frequently detected amongst the children exhibiting symptoms in comparison to asymptomatic children (p-value <0.001 for each pathogen) respectively, but *Giardia* was more isolated among the children without symptoms compared to symptomatic children (26.4% vs 18.9%, p-value <0.001).

The baseline demographic characteristics of the children are presented in S1 and S2 Tables. For this analysis, we included 7800 children. For the asymptomatic children, *Cryptosporidium* was detected in 843 (18.3%), *Giardia* in 3470 (75.2%), and *E. histolytica* in 299 (6.5%). *Cryptosporidium* and *E. histolytica* were more often (> 40%) detected among younger children (0–11 months), and *Giardia* was frequently detected among the older children (more than two years old) among whom, more than 40% were female. For the MSD cases, *Cryptosporidium* was detected in 1123 (35.2%), *Giardia* in 1786 (56%), and *E. histolytica* in 279 (8.8%). Around 55.6% of children in the 0–11 months age group were *Cryptosporidium*-positive. About 40% of children were from the older age group who were *Giardia*-positive. But *E. histolytica* was less common among the children who were 24 months old and above. Dysentery (42.3%) occurred substantially more frequently for the *E. histolytica*-positive children and fever (60.1%) were most observed in *Cryptosporidium* positive and *Giardia*-positive children where 55.56% *E. histolytica*-positive children were presented with fever.

In separate multiple linear regression models among symptomatic MSD children, after adjusting the potential covariates, we observed, that at ~60-day follow-up, *Cryptosporidium* infection was associated with lower HAZ (-0.17; 95% CI: -0.23, -0.11), WAZ (-0.25; 95% CI: -0.31, -0.19), and WHZ (-0.23; 95% CI: -0.30, -0.17) (Table 2) which was lower than expected during ~60 days follow up compared to the *Cryptosporidium* negative children. This finding

**Table 1. Baseline characteristics of the asymptomatic and symptomatic MSD children in South Asia and sub-Saharan Africa.**

| Characteristics<br>Total (n = 22,567) | | Asymptomatic Children<br>n = 13,128 | Symptomatic MSD children<br>n = 9,439 | *P value* |
|---|---|---|---|---|
| Age group | | | | |
| | 0-11m | 4,878 (37.2) | 4,030 (42.7) | ref |
| | 12-23m | 4,381 (33.4) | 3,205 (33.9) | 0.18 |
| | 24-59m | 3,870 (29.5) | 2,205 (23.4) | 0.01 |
| Gender (Girl) | | 5,651 (43.0) | 4,095 (43.4) | 0.39 |
| Baseline Anthropometry | | | | |
| | HAZ/LAZ¶ | -1.34± 1.31 | -1.34± 1.37 | 0.67 |
| | WAZ¶ | -1.08± 1.32 | -1.51± 1.39 | <0.001 |
| | WHZ¶ | -0.47± 1.49 | -1.06± 1.49 | <0.001 |
| Breastfeeding status | | | | |
| | Breastfed | 9,039 (68.9) | 6,741 (71.4) | ref |
| | Not breastfed | 4,090 (31.2) | 2,698 (28.6) | 0.13 |
| Primary caretaker's education | | | | |
| | Illiterate | 5,168 (39.4) | 4,016 (42.7) | 0.17 |
| | Literate | 7,935 (60.6) | 5,386 (57.3) | |
| Wealth index | | | | |
| | Poorest | 2,510 (19.1) | 2,027 (21.5) | ref |
| | lower middle | 2,590 (19.7) | 1,813 (19.2) | 0.13 |
| | Middle | 2,834 (21.6) | 1,993 (21.1) | 0.23 |
| | Upper middle | 2,522 (19.2) | 1,780 (18.9) | 0.28 |
| | Richest | 2,672 (20.4) | 1,821 (19.3) | 0.23 |
| The main source of drinking water | | | | |
| | Tube well water | 2,994 (22.8) | 1,675 (17.7) | ref |
| | Non-tube well water | 10,135 (77.2) | 7,765 (82.3) | <0.001 |
| Handwashing material | | | | |
| | With soap and water | 9,762 (74.4) | 7,131 (75.6) | ref |
| | Without soap | 3,365 (25.6) | 2,308 (24.5) | 0.64 |
| Handwashing practice | | | | |
| | Before nursing a child | 5,133 (39.1) | 3,683 (39.0) | 0.99 |
| | After cleaning a child who defecated | 6,189 (47.1) | 4,249 (45.0) | 0.79 |
| Available toilet facility | | | | |
| | sanitary/semi-sanitary | 12,289 (93.6) | 8,979 (95.1) | ref |
| | Non-sanitary | 840 (6.4) | 461 (4.9) | <0.001 |
| Common pathogens isolated | | | | |
| | *Cryptosporidium* | 843 (6.4) | 1,123 (11.9) | <0.001 |
| | *Giardia* | 3,470 (26.4) | 1,786 (18.9) | <0.001 |
| | *Entamoeba histolytica* | 299 (2.3) | 279 (2.9) | <0.001 |
| | ETEC | 975 (7.4) | 1,067 (11.3) | <0.001 |
| | *Campylobacter* | 1,561 (11.9) | 1,171 (12.4) | 0.63 |
| | EAEC | 2,655 (20.2) | 1,846 (19.6) | 0.33 |
| | Rotavirus | 509 (3.9) | 1,747 (18.5) | <0.001 |
| | *Shigella* | 231 (1.8) | 1,110 (11.8) | <0.001 |

¶ mean± SD (standard deviation); ETEC: Enterotoxigenic *E. coli*; EAEC: Enteroaggregative *E. coli*; HAZ/LAZ: height/length-for-age, WAZ: weight-for-age, and WHZ: weight-for-height z-scores

**Table 2. Among symptomatic children: association between enteric protozoan parasites infection and child anthropometric measurements: results of multiple linear regression modeling (dependent variables—HAZ/LAZ, WAZ, and WHZ) among the different age groups.**

| | Symptomatic MSD children | | | | | | | |
| | *Giardia* | | *Cryptosporidium* | | *E. histolytica* | | Presence of any one parasite | |
| | Coef. (95% CI) * | P value | Coef. (95% CI) * | P value | Coef. (95% CI) * | P value | Coef. (95% CI) * | P value |
|---|---|---|---|---|---|---|---|---|
| | Overall | | | | | | | |
| HAZ | -0.01(-0.06, 0.04) | 0.63 | -0.17(-0.23, -0.11) | <**0.001** | 0.03(-0.08, 0.14) | 0.62 | -0.08(-0.13, -0.04) | <**0.001** |
| WAZ | 0.02(-0.03, 0.07) | 0.50 | -0.25(-0.31, -0.19) | <**0.001** | 0.04(-0.07, 0.16) | 0.49 | -0.09(-0.13, -0.05) | <**0.001** |
| WHZ | 0.02(-0.04, 0.08) | 0.48 | -0.23(-0.30, -0.17) | <**0.001** | 0.02(-0.10, 0.15) | 0.70 | -0.08(-0.13, -0.03) | **0.001** |
| | 0–11 months | | | | | | | |
| HAZ | -0.004 (-0.10, 0.09) | 0.93 | -0.14(-0.22, -0.06) | <**0.001** | -0.06(-0.23, 0.11) | 0.48 | -0.12(-0.18, -0.05) | **0.001** |
| WAZ | -0.02(-0.13, 0.09) | 0.74 | -0.21(-0.30, -0.12) | <**0.001** | 0.08(-0.11, 0.28) | 0.40 | -0.14(-0.21, -0.06) | <**0.001** |
| WHZ | -0.1(-0.22, 0.03) | 0.13 | -0.21(-0.31, -0.11) | <**0.001** | 0.11(-0.10, 0.33) | 0.31 | -0.15(-0.24, -0.07) | <**0.001** |
| | 12–23 months | | | | | | | |
| HAZ | 0.02(-0.06, 0.10) | 0.69 | -0.19(-0.29, -0.09) | <**0.001** | 0.02(-0.17, 0.20) | 0.87 | -0.06(-0.13,0.01) | 0.09 |
| WAZ | 0.05(-0.03, 0.14) | 0.19 | -0.29(-0.39, -0.18) | <**0.001** | -0.05(-0.25, 0.14) | 0.58 | -0.07(-0.14, 0.003) | 0.06 |
| WHZ | 0.06(-0.02, 0.15) | 0.15 | -0.26(-0.37, -0.15) | <**0.001** | -0.06(-0.26, 0.15) | 0.58 | -0.05(-0.13,0.03) | 0.20 |
| | 24–59 months | | | | | | | |
| HAZ | -0.04(-0.12, 0.04) | 0.30 | -0.31(-0.47, -0.14) | <**0.001** | 0.22(-0.01, 0.44) | 0.0 | -0.06(-0.14,0.02) | 0.13 |
| WAZ | -0.02(-0.09, 0.06) | 0.66 | -0.21(-0.36, -0.06) | **0.005** | 0.09(-0.11, 0.29) | 0.39 | -0.06(-0.13,0.01) | 0.12 |
| WHZ | 0.02(-0.05, 0.10) | 0.57 | -0.07(-0.22, 0.08) | 0.37 | -0.06(-0.27, 0.15) | 0.59 | -0.02(-0.09,0.05) | 0.62 |

* Adjusted for sex, breastfeeding status, primary caretaker's education, WASH, wealth index, co-pathogens (ETEC, EAEC, *Shigella*, *Campylobacter*, and Rotavirus), site, and history of comorbidity (malaria, typhoid, pneumonia, diarrhea, dysentery) at day 60 follow up. Abbreviation: Coef.: coefficient, CI: confidence interval; HAZ/LAZ: height/length-for-age, WAZ: weight-for-age, and WHZ: weight-for-height z-scores; enteric protozoan parasites were detected from the stool sample during enrollment; Anthropometric measurements were taken during enrollment and after 60 days of enrollment (during the follow-up visit); Separate models were performed to see the association of enteric protozoan parasites infection with a child's height-for-age, weight-for-age, and weight-for-height z-scores for the symptomatic and asymptomatic infection

was consistent for nearly all stratified age groups. Among children who were positive for *Giardia* and *E. histolytica*, we did not find any significant associations with HAZ, WAZ, and WHZ in 60 days of follow-up. Children positive for any one of the enteric protozoan parasites (*Giardia* or *Cryptosporidium* or *E. histolytica*) showed a significant decrease in each child growth outcome among overall all children and among the 0–11 months age children, but not for the older children (12–23 months and 24–59 months) (Table 2).

Among asymptomatic children at ~60-day follow-up, *Cryptosporidium* infection was associated with lower WAZ (-0.15; 95% CI: -0.22, -0.09) and WHZ (-0.18; 95% CI -0.25, -0.12), which was lower than expected during follow up; this relationship did not exist for HAZ (-0.03; 95% CI -0.09, 0.04) compared to *Cryptosporidium* negative children. This finding was consistent for nearly all stratified age groups except for older children (24–59 months old). *Giardia* infection was associated with lower HAZ (-0.13; 95% CI: -0.17, -0.09) and WAZ (-0.07; 95% CI: -0.11, -0.04); but not WHZ (-0.02; 95% CI -0.06, 0.02) which was lower than expected in comparison to the *Giardia* negative children, these findings were also observed amongst the different age groups of children except for older children (24–59 months). No associations were found between *E. histolytica* and growth measures. Positivity for any one of the enteric protozoan parasites (*Giardia* or *Cryptosporidium* or *E. histolytica*) was associated with lower child growth outcomes among overall all children, but not for the older children (24–59 months). In 24–59 months, children, who were positive for any one of the enteric protozoan parasites had a

**Table 3. Among asymptomatic children: association between enteric protozoan parasites infection and child anthropometric measurements: results of multiple linear regression modeling (dependent variables—HAZ/LAZ, WAZ, and WHZ) among the different age groups.**

| | Asymptomatic children | | | | | | | |
| | Giardia | | Cryptosporidium | | E. histolytica | | Presence of any one parasite | |
| | Coef. (95% CI) * | P value | Coef. (95% CI) * | P value | Coef. (95% CI) * | P value | Coef. (95% CI) * | P value |
|---|---|---|---|---|---|---|---|---|
| | Overall | | | | | | | |
| HAZ | -0.13 (-0.17, -0.09) | <**0.001** | -0.03 (-0.09,0.04) | 0.40 | -0.04 (-0.15,0.06) | 0.42 | -0.12 (-0.15, -0.08) | <**0.001** |
| WAZ | -0.07 (-0.11, -0.04) | <**0.001** | -0.15 (-0.22, -0.09) | <**0.001** | -0.08 (-0.19,0.02) | 0.12 | -0.10 (-0.14, -0.07) | <**0.001** |
| WHZ | -0.02 (-0.06,0.02) | 0.36 | -0.18 (-0.25, -0.12) | <**0.001** | -0.07 (-0.18,0.05) | 0.27 | -0.06 (-0.10, -0.02) | **0.002** |
| | 0-11months | | | | | | | |
| HAZ | -0.14 (-0.22, -0.06) | **0.001** | -0.02 (-0.13,0.08) | 0.66 | -0.11 (-0.27,0.06) | 0.20 | -0.12 (-0.19, -0.05) | <**0.001** |
| WAZ | -0.13 (-0.22, -0.04) | **0.003** | -0.15 (-0.26, -0.04) | **0.007** | -0.03 (-0.2,0.14) | 0.73 | -0.16 (-0.23, -0.09) | <**0.001** |
| WHZ | -0.13 (-0.23, -0.03) | **0.008** | -0.19 (-0.32, -0.07) | **0.002** | 0.08 (-0.11,0.27) | 0.42 | -0.15(-0.23, -0.07) | <**0.001** |
| | 12–23 months | | | | | | | |
| HAZ | -0.17 (-0.23, -0.11) | <**0.001** | -0.03 (-0.14,0.08) | 0.65 | 0.09 (-0.11,0.28) | 0.38 | -0.14 (-0.2, -0.08) | <**0.001** |
| WAZ | -0.10 (-0.16, -0.04) | **0.002** | -0.21 (-0.33, -0.1) | <**0.001** | -0.12 (-0.32,0.07) | 0.22 | -0.12 (-0.18, -0.07) | <**0.001** |
| WHZ | -0.02 (-0.09,0.04) | 0.51 | -0.27 (-0.39, -0.14) | <**0.001** | -0.22 (-0.43, -00004) | 0.05 | -0.08 (-0.14, -0.01) | **0.017** |
| | 24–59 months | | | | | | | |
| HAZ | -0.09 (-0.15, -0.04) | **0.001** | -0.08 (-0.20,0.03) | 0.15 | -0.10 (-0.31,0.10) | 0.32 | -0.10 (-0.16, -0.05) | <**0.001** |
| WAZ | -0.02 (-0.07,0.02) | 0.33 | -0.04 (-0.15,0.06) | 0.40 | -0.17 (-0.35,0.01) | 0.07 | -0.04 (-0.08,0.01) | 0.16 |
| WHZ | 0.05 (-0.004,0.10) | 0.07 | 0.02 (-0.09,0.12) | 0.77 | -0.19 (-0.37,0.003) | 0.05 | 0.04 (-0.01,0.09) | 0.10 |

* Adjusted for sex, breastfeeding status, primary caretaker's education, WASH, wealth index, co-pathogens (ETEC, EAEC, *Shigella*, *Campylobacter*, and Rotavirus), site, and history of comorbidity (malaria, typhoid, pneumonia, diarrhea, dysentery) at day 60 follow up. Abbreviation: Coef.: coefficient, CI: confidence interval; HAZ/LAZ: height/length-for-age, WAZ: weight-for-age, and WHZ: weight-for-height z-scores; enteric protozoan parasites were detected from the stool sample during enrollment; Anthropometric measurements were taken during enrollment and after 60 days of enrollment (during the follow-up visit); Separate models were performed to see the association of enteric protozoan parasites infection with a child's height-for-age, weight-for-age, and weight-for-height z-scores for the symptomatic and asymptomatic infection

significant decrease in HAZ (Table 3) which was lower than expected during follow-up compared to the children who were negative for any protozoal infection.

To gain further insight into the association of protozoan parasite detection with child growth, we explored the role of co-pathogen colonization. We observed an association (p = 0.01) and WHZ (β: -0.21; 95% CI: -0.35, -0.06; p <0.001); on the other hand, co-infection with both *Giardia* and *E. histolytica* was associated with higher growth than the expected HAZ (β: 0.32; 95% CI: 0.07, 0.58; p = 0.01) and WAZ (β: 0.30; 95% CI: 0.03, 0.56; p = <0.001) among the children exhibiting symptoms. No association was found between the co-colonization of *Cryptosporidium* and *E. histolytica* on diarrhea symptoms (Table 4). We report on the prevalence of multiple infections in S3 Table and compared enteric protozoan parasite co-infections between asymptomatic and symptomatic MSD children in S4 Table.

Most co-infections did not reveal an association with growth measures (Table 4). For symptomatic MSD children, exceptions included negative associations with growth when children were positive for *Cryptosporidium* with *E. Histolytica* negative and *Cryptosporidium* regardless of *Giardia* status (I.e., both Giardia positive and negative).These relationship with lower growth when children had co-infection with both *Cryptosporidium* and *E. histolytica* also held for asymptomatic MSD children, but among these children, negative associations with growth existed when children were positive for *Giardia* regardless of *Cryptosporidium* status. We explored the age-stratified association between parasitic co-infections and child anthropometry. In S5A and S5B Table.

**Table 4. Associations between enteric protozoan parasites co-infections and child anthropometric measurements among under 5 asymptomatic and symptomatic MSD children in South Asia and sub-Saharan Africa using multiple linear regression modeling (Dependent variables: HAZ/LAZ, WAZ, and WHZ).**

| | Symptomatic MSD children | | | | | |
|---|---|---|---|---|---|---|
| | Coef. (95% CI) * | P value | Coef. (95% CI) * | P value | Coef. (95% CI) * | P value |
| | *Crypto (+) and E. histolytica (+* | | *Crypto (+) and E. histolytica (-)* | | *Crypto (-) and E. histolytica (+)* | |
| HAZ | -0.02 (-0.29, 0.26) | 0.90 | **-0.17 (-0.23, -0.11)** | **<0.001** | 0.05 (-0.07,0.17) | 0.44 |
| WAZ | 0.04 (-0.25, 0.34) | 0.78 | **-0.25 (-0.31, -0.19)** | **<0.001** | 0.05 (-0.07,0.18) | 0.42 |
| WHZ | 0.12 (-0.2, 0.44) | 0.45 | **-0.24 (-0.31, -0.17)** | **<0.001** | 0.02 (-0.12,0.15) | 0.80 |
| | *Crypto (+) and Giardia (+)* | | *Crypto (+) and Giardia (-)* | | *Crypto (-) and Giardia (+)* | |
| HAZ | -0.09 (-0.21, 0.03) | 0.13 | **-0.17 (-0.24, -0.11)** | **<0.001** | 0.01 (-0.05, 0.06) | 0.79 |
| WAZ | **-0.18 (-0.31, -0.05)** | **0.01** | **-0.23 (-0.3, -0.16)** | **<0.001** | 0.06 (0.001, 0.11) | 0.05 |
| WHZ | **-0.21 (-0.35, -0.06)** | **<0.001** | **-0.21 (-0.28, -0.13)** | **<0.001** | 0.06 (0.004, 0.12) | 0.04 |
| | *Giardia (+) and E. histolytica (+)* | | *Giardia (+) and E. histolytica (-)* | | *Giardia (-) and E. histolytica (+)* | |
| HAZ | **0.32 (0.07, 0.58)** | **0.01** | -0.03 (-0.08, 0.02) | 0.24 | -0.05 (-0.17, 0.08) | 0.46 |
| WAZ | **0.30 (0.03, 0.56)** | **0.03** | -0.003 (-0.06, 0.05) | 0.90 | -0.03 (-0.16, 0.09) | 0.60 |
| WHZ | 0.18 (-0.11, 0.48) | 0.22 | 0.003 (-0.05, 0.06) | 0.90 | -0.03 (-0.16, 0.11) | 0.69 |
| | Asymptomatic children | | | | | |
| | Coef. (95% CI) * | P value | Coef. (95% CI) * | P value | Coef. (95% CI) * | P value |
| | *Crypto (+) and E. histolytica (+)* | | *Crypto (+) and E. histolytica (-)* | | *Crypto (-) and E. histolytica (+)* | |
| HAZ | 0.08 (-0.28, 0.45) | 0.66 | -0.04 (-0.10, 0.02) | 0.23 | -0.05 (-0.16, 0.06) | 0.34 |
| WAZ | -0.22 (-0.6, 0.16) | 0.25 | **-0.15 (-0.22, -0.09)** | **<0.001** | -0.07 (-0.18, 0.04) | 0.24 |
| WHZ | -0.32 (-0.74, 0.09) | 0.13 | **-0.18 (-0.25, -0.11)** | **<0.001** | -0.04 (-0.16, 0.08) | 0.54 |
| | *Crypto (+) and Giardia (+)* | | *Crypto (+) and Giardia (-* | | *Crypto (-) and Giardia (+)* | |
| HAZ | -0.03 (-0.14, 0.09) | 0.62 | -0.01 (-0.09, 0.07) | 0.81 | **-0.13 (-0.17, -0.09)** | **<0.001** |
| WAZ | -0.08 (-0.19, 0.03) | 0.17 | **-0.16 (-0.24, -0.08)** | **<0.001** | **-0.07 (-0.10, -0.03)** | **<0.001** |
| WHZ | -0.08 (-0.20, 0.03) | 0.17 | **-0.20 (-0.28, -0.12)** | **<0.001** | -0.01 (-0.05, 0.03) | 0.72 |
| | *Giardia (+) and E. histolytica (+)* | | *Giardia (+) and E. histolytica (-)* | | *Giardia (-) and E. histolytica (+)* | |
| HAZ | -0.02 (-0.23, 0.19) | 0.87 | -0.03 (-0.08, 0.02) | 0.24 | -0.05 (-0.17, 0.08) | 0.46 |
| WAZ | -0.16 (-0.36, 0.05) | 0.13 | -0.003 (-0.06, 0.05) | 0.90 | -0.03 (-0.16, 0.09) | 0.60 |
| WHZ | -0.2 2(-0.44, -0.004) | 0.05 | 0.003 (-0.05, 0.06) | 0.90 | -0.03 (-0.16, 0.11) | 0.69 |

* Adjusted for sex, breastfeeding status, primary caretaker's education, WASH, wealth index, co-pathogens (ETEC, EAEC, *Shigella*, *Campylobacter*, and Rotavirus), site, and history of comorbidity (malaria, typhoid, pneumonia, diarrhea, dysentery) at day 60 follow up. Abbreviation: Coef.: coefficient, CI: confidence interval; HAZ/LAZ: height/length-for-age, WAZ: weight-for-age, and WHZ: weight-for-height z-scores; enteric protozoan parasites were detected from the stool sample during enrollment; Anthropometric measurements were taken during enrollment and after 60 days of enrollment (during the follow-up visit); Separate models were performed to see the association of enteric protozoan parasites infection with a child's height-for-age, weight-for-age, and weight-for-height z-scores for the symptomatic and asymptomatic infection

## Discussion

To our knowledge, this is the first study to investigate the associations between the detection of enteric protozoan parasites with child growth; we included data from children in multiple countries ranging from birth until 5 years of age with and without diarrhea. We found compelling evidence that infection with *Cryptosporidium* impacts WAZ and WHZ among children enrolled as both symptomatic and asymptomatic; but association with HAZ was only for symptomatic children. Associations with *Giardia* infection were found for asymptomatic children only for HAZ and WAZ, but not WHZ, and no associations were found among symptomatic children. No associations were found among children positive for *E. histolytica*, perhaps owing to limited power from low prevalence. Associations with any infection were

found among those in both the symptomatic and asymptomatic groups for all three growth categories, though the strength of that association differed within age groups.

Our results have significant implications for the prevention and management of asymptomatic intestinal protozoan parasitic infections in developing countries. It is widely known that children in developing countries are found to be infected with intestinal protozoan parasitic infections and often do not have diarrhea [16]. In some cases, enteric parasites result in self-limiting diarrhea; thus, typically, infected individuals are managed supportively, and not prescribed antiprotozoal treatment. The findings of this analysis suggest that this decision tree should be re-evaluated for under 5 children, given the substantial potential impact on growth faltering/detrimental effect on the nutrition status of asymptomatic intestinal protozoan parasitic infected children.

Our analysis showed that asymptomatic *Giardia* infection was significantly associated with both HAZ and WAZ, indicating poor linear growth and acute energy deficiency in affected children. However, there was no observed impact of *Giardia* on child growth among symptomatic children. The effect of symptomatic *Giardia*-associated illness on child growth remains controversial. A previous study noted that *Giardia*-infected children in Guatemala had poorer weight gain than those without this infection, but only in the second year of life; no effect was seen in the first or third years [17]. In another study in The Gambia, it was found that *Giardia* was not a major cause of the poor growth of the children [18]. Earlier studies have suggested that systemic invasion by enteric parasites could contribute to growth retardation by causing acute blood loss and depletion of body protein stores [4]. *Giardia* evades host inflammatory responses and may affect epithelial cells through various mechanisms, including producing antioxidants, cleaving interleukin-8, depleting arginine via arginine deiminase, shifting variant surface protein expression, cell-cycle arrest, proliferation impairment, tight-junction disruption, and apoptosis induction [19]. Enteric parasites can cause apical junction complex breakdown leading to intestinal permeability and allowing microbial and food antigens to enter the sub-epithelial compartment, as well as lactose malabsorption in affected individuals [20]. The study from Kerman, central Iran demonstrated that symptomatic giardiasis was characterized by significantly elevated serum levels of the TH1 cytokine IFN-γ compared to healthy controls, while asymptomatic human subjects and healthy controls had comparable levels of serum IFN-γ, and that *Giardia* genotype AI infection was associated with significantly elevated levels of serum IFN-γ and IL-10, but not IL-5, compared to healthy controls [21]. These findings may demonstrate the roles of both host and parasite factors in the determination of the growth outcome of *Giardia* infections, regardless of diarrheal symptoms.

Our data showed that acute malnutrition, as measured by WHZ, was significantly associated with subsequent growth faltering, which could be attributed to severe intestinal injury and absorptive capacity, and several studies have suggested that ponderal growth may precede linear growth as weight loss reflects a lack of available nutrients required for sustaining linear growth [22]. Our study suggests that the increased risks of severe linear growth faltering in asymptomatic children may be associated with higher rates of subsequent diarrhea episodes during the follow-up period, as previous research has reported a higher incidence of diarrhea in acutely malnourished children [23].

Contradictory findings regarding the protective role of *Giardia* in early childhood have been reported based on single pathogen colonization data from a community-based intervention study [24]. Children with *Giardia* detected in their duodenal aspirates have shown increased intestinal permeability, which may contribute to stunting and other forms of malnutrition, independent of chronic inflammation of the gut tissue. Giardiasis also has shown a minimal association with markers of intestinal inflammation, further suggesting a non-inflammatory mechanism [25], causing fewer symptoms. Our study found no significant growth

faltering in children with symptomatic *Giardia* infection, possibly due to the activation of innate immunity which can protect against other enteric illnesses [26]. The reduced occurrence of *Giardia* in diarrheal stools suggests it may act as a barrier against other pathogens, but we propose an alternative interpretation: colonized *Giardia* is flushed out during acute diarrheal episodes caused by other pathogens, and this does not necessarily indicate protection against undernutrition.

Children with symptomatic and asymptomatic *Cryptosporidium* infections had significantly more growth faltering compared to those without the infection, possibly due to malabsorption of nutrients and intestinal inflammation caused by the parasite. Intestinal inflammation is known to impair growth and development, as evidenced by elevated inflammatory markers such as C-reactive protein (CRP) and interleukin-6 (IL-6) be elevated in children with giardiasis and a link between chronic inflammation and stunting [27]. *Cryptosporidium* causes inflammation and damage to the gut lining, leading to malabsorption and malnutrition. Asymptomatic *Cryptosporidium* infections may not affect long-term growth in children due to the short duration and lower parasite burden, which are often cleared by the body's immune system.

Symptomatic *Cryptosporidium* strongly correlates with impaired HAZ in all age groups, likely due to its activation of the NF-kβ pathway and subsequent chronic inflammation and enteropathy. Linear growth faltering risk decreases as children age, with the highest risk before 12 months, in line with previous research [28]. These findings suggest the importance of early detection and treatment of *Cryptosporidium* to prevent long-term growth impairment in children. The risk of linear growth faltering decreases as children age, with the highest risk occurring before 12 months. In children with diarrhea presentation, the loss of height-for-age may have more health consequences than in asymptomatic children with *Cryptosporidium*, consistent with previous studies noting dysentery or specific pathogens known to cause dysentery to be associated with risks for linear growth faltering [29]. *Cryptosporidium* was associated with a greater decline in linear growth in children under 2 years with MSD, corroborating and expanding previous studies suggesting that *Cryptosporidium* in infancy imparts a lasting adverse effect on linear growth, including less weight gain in the first month of infection [9].

Our study did not find any significant association between *E. histolytica* infection and differences in child anthropometry. We have no clear explanation for this null association, but one possibility is the low prevalence's restricted power. However, the effects of the infection may be temporary, and our results may underestimate the true association due to oral rehydration and anti-amebic/antibiotic therapy provided to symptomatic children. Previous studies suggest that *E. histolytica* infection could contribute to growth retardation through acute blood loss and depletion of body protein stores [30]. In a study of under 2 children from GEMS, dysentery was more likely to be treated with antibiotics than other enteric infections, potentially contributing to a growth-promoting effect on length and weight, although the impact of antibiotic treatment on growth in children with enteric infections remains unclear [31]. Moreover, *Giardia* and *Cryptosporidium* colonize the duodenum, jejunum, and ileum and are the most common sites for nutrients absorption; but *E. histolytica* colonizes the colon not in the small intestine [32], this might be another provable explanation of our findings.

Our analysis revealed that child growth faltering attributed risk varies among the three enteric protozoan parasites, which could be related to dehydration status and severity of intestinal injury. Ponderal growth may precede linear growth, as weight loss may indicate a lack of nutrients needed for sustained linear growth, according to previous studies [22]. Our study suggests that nutritional interventions or antiprotozoal treatment guidelines may prevent childhood chronic malnutrition due to enteric parasite infection. However, further research is

necessary to understand the impact of cumulative protozoan colonization on gut homeostasis and its correlation with poor growth in children under five.

Based on the analysis of data regarding single pathogen colonization, contradictory findings have emerged regarding the potential protective role of *Giardia*, particularly during early childhood [16, 26]. In animal model experiments utilizing protein-deficient mice, it has been observed that microbial adaptations to undernutrition, in conjunction with the cumulative impact of entomopathogens, significantly influence host growth, gut immune response, and metabolism [33]. In the context of *Giardia* and enteroaggregative *Escherichia coli* co-colonization, it has been observed that *Giardia* can effectively evade microbiota-mediated pathogen clearance under conditions of protein malnutrition. This evasion mechanism leads to the promotion of growth impairment and significant alterations in the abundance of small intestinal 16S, as well as mucosal immunity. These changes converge with the metabolic responses to exacerbate host growth impairment. [34]. Their data model the effect of early-life cumulative enteropathogen exposures on the disruption of intestinal immunity and host metabolism during crucial developmental periods [34]. We observed similar trends in our study of children where *Cryptosporidium* detection in the presence of *Giardia* had a negative impact on the child's growth, commonly among 24–59 months old diarrhoeal children. Similarly, simultaneously positive for *Giardia* and *E. histolytica* had poor growth in 12–23 months old children who were asymptomatic. As intestinal microbiota develops during the early months of life, this protection might provide the critical window of susceptibility. However, Cryptosporidium causes persistent diarrhea in young children in the LMICs. Although it has been suggested that the effect of infections on nutrition is usually transient because of catch-up growth, another community study from West Africa suggested that cryptosporidiosis in infancy has a permanent effect on growth [35]. Hence, further studies are required on the role of gut microbiota and enteropathogen co-colonization.

Our study was limited by the lack of data on maternal BMI, gestational age, birth weight, serum micronutrient levels, and enteric inflammatory biomarkers of the children. Additionally, only short-term impacts (60 days) were assessed, leaving open the possibility of subsequent catch-up growth. We could not detect the effect of the human immunodeficiency virus (HIV) as HIV data in the children's population were not readily available for South Asia. Our study had a large, randomly sampled population and high-quality lab procedures. We examined the relationship between intestinal protozoan parasitic infection and growth faltering in children under five across seven sites, with a single follow-up visit allowing us to analyze growth outcomes during the at-risk period. This enteric protozoan parasitic infection data can be highly valuable in advancing our understanding of enteric infections and their impact on child health and development. Our analysis includes information on co-infections and co-pathogens, providing insights into how multiple parasites or pathogens interact and influence each other's effects on child health. Understanding these interactions can have implications for diagnosis, treatment, and public health strategies. Our analysis findings can be used to assess the effectiveness of interventions targeting parasitic infections, such as WASH improvements, deworming programs, and nutrition interventions.

## Conclusion

Enteric protozoan parasites are associated with lower growth in young children, regardless of diarrheal symptoms, with *Cryptosporidium* and *Giardia* being associated with growth shortfalls among asymptomatic children. Future studies are needed to determine the impact of treating or preventing asymptomatic enteric protozoan parasite-associated illness on childhood malnutrition, and regular monitoring for parasitic infection should be included in

nutritional assessments in endemic regions. Research to develop effective drugs and vaccines for enteric protozoan parasitic infections is crucial to prevent childhood malnutrition.

## Supporting information

**S1 Table. Baseline characteristics of the asymptomatic children having stool positive for enteric protozoan parasites (*Cryptosporidium*, *Giardia*, and *Entamoeba histolytica*) in seven sites of GEMS.**
(DOCX)

**S2 Table. Baseline characteristics of the symptomatic children having stool positive for enteric protozoan parasites (*Cryptosporidium*, *Giardia*, and *Entamoeba histolytica*) in seven sites of GEMS.**
(DOCX)

**S3 Table. Children having more than one enteric protozoan parasite**
(DOCX)

**S4 Table. Comparing enteric protozoan parasite co-infections detected among asymptomatic and symptomatic MSD children in GEMS (n = 22,567).**
(DOCX)

**S5 Table. Age-specific association between enteric protozoan parasites co-infections and child anthropometric measurements among symptomatic MSD and asymptomatic children on growth measures among under 5 children in South Asia and sub-Saharan Africa using multiple linear regression modeling (Dependent variables: HAZ/LAZ, WAZ, and WHZ). a**. Age-specific association between enteric protozoan parasites co-infections and child anthropometric measurements among symptomatic MSD children on growth measures among under 5 children in South Asia and sub-Saharan Africa using multiple linear regression modeling (Dependent variables: HAZ/LAZ, WAZ, and WHZ). **b**. Age-specific association between enteric protozoan parasites co-infections and child anthropometric measurements among asymptomatic children on growth measures among under 5 children in South Asia and sub-Saharan Africa using multiple linear regression modeling (Dependent variables: HAZ/LAZ, WAZ, and WHZ).
(DOCX)

## Acknowledgments

The authors are grateful to GEMS staff, parents, and children for their contributions. GEMS research protocol was funded by the Bill & Melinda Gates Foundation. We acknowledge the contribution of icddr,b's core donors including the Government of the People's Republic of Bangladesh, Global Affairs Canada (GAC), Canada; Swedish International Development Cooperation Agency and Foreign, Commonwealth and Development Office (FCDO), UK for their continuous support and commitment to icddr,b's research efforts.

## Author Contributions

**Conceptualization:** Rina Das, Tahmeed Ahmed, Robert F. Breiman, A. S. G. Faruque.

**Data curation:** Rina Das, Md. Ahshanul Haque.

**Formal analysis:** Rina Das.

**Funding acquisition:** Rina Das, A. S. G. Faruque.

**Investigation:** Myron M. Levine, Karen L. Kotloff, Dilruba Nasrin, M. Jahangir Hossain, Dipika Sur, Robert F. Breiman.

**Methodology:** Rina Das, A. S. G. Faruque.

**Project administration:** Rina Das, A. S. G. Faruque.

**Resources:** A. S. G. Faruque.

**Supervision:** Robert F. Breiman, Matthew C. Freeman, A. S. G. Faruque.

**Validation:** Robert F. Breiman, Matthew C. Freeman, A. S. G. Faruque.

**Visualization:** Rina Das.

**Writing – original draft:** Rina Das.

**Writing – review & editing:** Rina Das, Parag Palit, Md. Ahshanul Haque, Myron M. Levine, Karen L. Kotloff, Dilruba Nasrin, M. Jahangir Hossain, Dipika Sur, Tahmeed Ahmed, Robert F. Breiman, Matthew C. Freeman, A. S. G. Faruque.

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
