## [Decision Letter · Decision Letter 0]

27 Jul 2023

Dear Dr. Das,

Thank you very much for submitting your manuscript "Symptomatic and asymptomatic enteric protozoan parasitic infection and their association with growth in young under 5 children in South Asia and sub-Saharan Africa" for consideration at PLOS Neglected Tropical Diseases. As with all papers reviewed by the journal, your manuscript was reviewed by members of the editorial board and by several independent reviewers. In light of the reviews (below this email), we would like to invite the resubmission of a significantly-revised version that takes into account the reviewers' comments. 

Dear Rina Das and co-authors,

Thank you for your submission. We apologize for the lag time in responding. This important work is of high interest, and we invite your responses to the reviews. 

Please address all reviewer comments, and particularly those raising needs to improve clarity in the analysis and presentation of the data. I agree that inclusion of qPCR data, if available, would enhance the manuscript, particularly if there are quantitative relationships between growth decrements and parasite burdens (though qPCR data is not a requirement). Please clearly address concerns raised by reviewer #1 regarding data ownership and whether these data agreement structures in GEMS require inclusion of co-authors from every study site. 

In addition to specific references needed in the discussion as raised by Reviewer #2, please perform an updated literature review in your revision and include other studies of Giardia/Cryptosporidium growth faltering that have been published in the interim.

We cannot make any decision about publication until we have seen the revised manuscript and your response to the reviewers' comments. Your revised manuscript is also likely to be sent to reviewers for further evaluation.

Sincerely,

Luther A Bartelt

Academic Editor

Charles Jaffe

Section Editor

Dear Rina Das and co-authors,

Thank you for your submission. We apologize for the lag time in responding. This important work is of high interest, and we invite your responses to the reviews. 

Please address all reviewer comments, and particularly those raising needs to improve clarity in the analysis and presentation of the data. I agree that inclusion of qPCR data, if available, would enhance the manuscript, particularly if there are quantitative relationships between growth decrements and parasite burdens (though qPCR data is not a requirement). Please clearly address concerns raised by reviewer #1 regarding data ownership and whether these data agreement structures in GEMS require inclusion of co-authors from every study site. 

In addition to specific references needed in the discussion as raised by Reviewer #2, please perform an updated literature review in your revision and include other studies of Giardia/Cryptosporidium growth faltering that have been published in the interim.

Reviewer's Responses to Questions

**Key Review Criteria Required for Acceptance?**

**Methods**

-Are the objectives of the study clearly articulated with a clear testable hypothesis stated?

-Is the study design appropriate to address the stated objectives?

-Is the population clearly described and appropriate for the hypothesis being tested?

-Is the sample size sufficient to ensure adequate power to address the hypothesis being tested?

-Were correct statistical analysis used to support conclusions?

-Are there concerns about ethical or regulatory requirements being met?

Reviewer #1: -Are the objectives of the study clearly articulated with a clear testable hypothesis stated?

yes

-Is the study design appropriate to address the stated objectives?

Yes

-Is the population clearly described and appropriate for the hypothesis being tested?

Yes GEMS samples are taken for testing hypothesis of Giardia / parasite infection on growth faltering

-Is the sample size sufficient to ensure adequate power to address the hypothesis being tested?

7800 children's data from GEMS being used.

-Were correct statistical analysis used to support conclusions?

Yes appropriate for growth data

-Are there concerns about ethical or regulatory requirements being met?

Gems is a multi-site project and ownership of data is with PIs of site. It requires ERC exemption, or all site PIs should be informed.

Reviewer #2: Authors have employed pathogen results detected by ELISA. What was the reasoning not to employ the detection results from the later reanalysis by molecular techniques? Would this not have given more accurate results, especially for the low-grade protracted infections that both Giardia and Cryptosporidium can cause? And thereby perhaps a better corrective balancing for all the co-pathogen variables in the regression model?

Anthropometric measures were done at two timepoints. It is commendable that both measures are available and included in the analysis, and that measures at enrolment were done after rehydration in case of MSD. This makes possible an evaluation of anthropometry status both at enrolment and at follow-up, and could help solve the puzzle regarding the degree to which malnourished children more often acquire protozoan infections versus the hypothesis that protozoan infections cause growth faltering, and how these factors may interact. 

Perhaps it is just my poor understanding of the statistical model, but I am bit confused about the results given in table 1 and 2. It is stated that they are “association between enteric protozoan parasites infection and child HAZ/LAZ, WAZ, and WHZ”. I then wonder; Are they expressing the deviation in score for each given protozoan pathogen from the expected increase in anthropometric scores in the whole cohort?

Authors also state that “Giardia infection was associated with lower HAZ (-0.13; 95% CI: -0.17, -0.09…». Does it actually mean a “lower than expected” or “reduced” HAZ score? Or the adjusted difference in the HAZ score from each child’s baseline 2 months previously, as if no growth was expected in that period? 

I think the authors need to communicate clearer how the results convey the difference between enrollment and follow-up for children found positive for these pathogens. Please give a clear description of the metric you use so that also non-epidemiologists can understand it. In Nasrin et al 2022 the data is given as “Difference in ΔHAZ (95% CI)”. Is that what is actually meant?

I would also strongly recommend to provide an example of what the coefficients given actually indicate. For example; what does a HAZ coefficient of -0.13 mean in height loss in cm for a child aged one year?

Important co-pathogens other than protozoans were included in the GLL model as co-variates, while presence of protozoans were explanatory variables. Can the authors clarify how co-infections with two or three protozoans (which are not uncommon) were handled by this model? For example, were any interactions between these explanatory variables tested? The Nasrin et al 2022 included a multiple-pathogen model estimate. Is that the same that was used in the present analyses?

**Results**

-Does the analysis presented match the analysis plan?

-Are the results clearly and completely presented?

-Are the figures (Tables, Images) of sufficient quality for clarity?

Reviewer #1: -Does the analysis presented match the analysis plan?

Analyses plan is adequate.

-Are the results clearly and completely presented?

Yes

-Are the figures (Tables, Images) of sufficient quality for clarity?

Need to submit better quality pictures

Reviewer #2: Authors start by giving prevalences of protozoans at different sites. In supplementary table 1 and 2 are given the number of positives for each of the three protozoan pathogens. Obviously, there must have been many samples positive for more than one protozoan pathogen. How did the authors arrive at the numbers in these tables, that are without protozoan co-infections, but still add up to 100%. How was co-infections dealt with. This could be better explained in the methods chapter, and some measure of the overlap given.

Ideally anthropometry follow up data should be given also for the co-infections as has been shown to give interesting results in a recent paper by Kabir et al. 2023 https://doi.org/10.3389/fnut.2022.1081833 As we know there are interactions here the present manuscript might miss interesting findings by seemingly presenting only single infection analysis.

Figure 1 gives a good overview of the participants, and show low attrition rate for follow up anthropometry measurement.

Line 119: Were data from infants who were negative for protozoan parasites not included in the analysis? Would they not have provided important background data, especially for the asymptomatic group?

Baseline anthropometric data are given in the supplementary tables 1 and 2 but not commented upon. Were there no significant baseline differences between protozoan pathogens? I think these results are also an important finding in such a large dataset, and should be stated, and possibly discussed later, as several studies to date have drawn (often uncertain) conclusions about malnutrition and protozoan infection based on such one-time measures. Asymptomatic controls seem to have higher WAZ and WHZ scores than symptomatic infants. I definitely would want a column showing the baseline data for the total population, including anthropometric data, in these two tables.

A short translation of the anthropometric measures into prevalence of MAM and SAM (based on WHZ) and stunting (based on HAZ) in the text would be helpful to put the study population nutritional status into perspective.

**Conclusions**

-Are the conclusions supported by the data presented?

-Are the limitations of analysis clearly described?

-Do the authors discuss how these data can be helpful to advance our understanding of the topic under study?

-Is public health relevance addressed?

Reviewer #1: -Are the conclusions supported by the data presented?

Yes 

-Are the limitations of analysis clearly described?

Not mentioned

-Do the authors discuss how these data can be helpful to advance our understanding of the topic under study?

not really clear. 

-Is public health relevance addressed?

Targeting enteric parasites in young children is mentioned

Reviewer #2: The discussion needs to touch upon repeated results are brought forward here and how they relate to the previous publication by Nasrin et al 2022. For example were Cryptosporidium in the symptomatic infants was published in that paper.

It would be good to include the reasoning for analysing single pathogens only, and not include results for protozoan co-infections. The authors have instead given estimates for “Presence of any one parasite” which I think gives less clinically interesting information. Please discuss the presence of protozoan co-infections in LMIC and preferably also include such co-infection analyses in this manuscript.

The authors refer to several studies of growth faltering in Giardia and Cryptosporidium. I think there are at least two studies that also should be included in this discussion, the early study by Mølbak et al and the recent study by Kapir et al. 

Mølbak K, Andersen M, Aaby P, Højlyng N, Jakobsen M, Sodemann M, et al. Cryptosporidium infection in infancy as a cause of malnutrition: a community study from Guinea-Bissau, west Africa. Am J Clin Nutr. 1997; 65(1):149–52. https://doi.org/10.1093/ajcn/65.1.149 PMID: 8988927

Kabir F et al. Impact of enteropathogens on faltering growth in a resource-limited setting. Front Nutr. 2023 Jan 10;9:1081833. doi: 10.3389/fnut.2022.1081833. PMID: 36704796 

Arguments relating to dysentery in lines 314 and onwards would be better placed in the section discussing E.histolytica. I think here the authors may also consider mentioning the effect of the different intestinal segments affected by each protozoan infection to explain the growth faltering seen in Giardia and Crypto, but not in E.histolytica. G and C infects the small intestine with its nutrient absorption role, while E.h is mainly a colonic pathogen.

**Editorial and Data Presentation Modifications?**

Reviewer #1: This is a secondary analyses of primary GEMS data. It has been published elsewhere in detail, however this is a new finding of Giardia being negatively associated in asymptomatic children. While previously Giardia has shown to be protective in linear growth faltering. The definition of asymptomatic /controls should be clearly defined. Giardia was measured through ELISA and not through microscopy. q-RT PCR data is also available for GEMS pathogens, but it was not included in this manuscript. It is important to understand how data was accessed for this manuscript from all sites, for instance Pakistan Gems data was used but there is no author on this manuscript. If it is a longitudinal data set which is not clearly mentioned here then probably Mixed effects model is more appropriate. For secondary data analyses, I would us robust analyses plan that has not been previously used rather than simple regression analyses. Outcome measure is not clearly defined, the impact of entomopathogen was observed within two months of life, this time frame is very short to observe long term effects of pathogen on growth.

Reviewer #2: Some Minor specific points

Line 34 in abstract, missing “-“ before 0.09 in the confidence interval.

Line 93-95: Incomprehensible sentence, please revise.

Line 197: The formulation “More than 55.6% of the …” seems a bit odd. Was it 55.6% or a unspecified higher percentage?

Line 240 Remove extra d in “investigated”

Line 247 reomve extra “both”

**Summary and General Comments**

Reviewer #1: 1- title should be modified - may consider omitting "young" 

2- under 5 should be changed to "five"

3- in abstract add p value with B estimates

4- Older children term was used. please define "older"

5- line # 189 " this needs further elaboration if asymptomatic were healthy children?"

6- (+) sign should be written as positive

7- In figure 3 title healthy term was used. it is confusing whether asymptomatic were healthy as well. 

8- Title of table should be revised. use anthropometric measurement rather than WHZ/LAZ/WAZ

9- Use either of LAZ or HAZ

10-Line # 242: Typo error: WHZ instead of HAZ

11- Figure 1: Gems control was mentioned but in whole manuscript control term was not used

Reviewer #2: The authors have examined a publicly available dataset from the GEMS study with regard to intestinal protozoans in symptomatic and asymptomatic infants below 5 years of age, and their association with anthropometric outcomes. They find significant associations between growth parameters and asymptomatic Giardia infection and for both asymptomatic and symptomatic Cryptosporidium infection, but not with E. histolytica. It is a valuable analysis and needed analysis of available data.

I have two major comments. How were protozoan co-infections dealt within the analysis and can the data say more about these? The other relates to clarity and communication of what the results actually mean.

I think there is potential for a much clearer and better analysis and communication of these data. Structure could be improved by consistently starting with either symptomatic of asymptomatic infection in all sections of the manuscript.

I would recommend authors to put more emphasis on the fact that data presented here are including a post-infection follow-up period, which is very valuable. I suggest they bring it into the title so that it will be easily understood what this paper is about. For example “ … association with subsequent growth parameters in children below 5 years …” not just “… association with growth in young under 5 children …”

In the introduction, please make a clear and comprehensive delineation of what was reported in the previous publication by Nasrin et al 2022 regarding HAZ in MSD cases in the GEMS data, and what new analyses are to be made in the present analysis.

PLOS authors have the option to publish the peer review history of their article (what does this mean?). If published, this will include your full peer review and any attached files.

Reviewer #1: No

Reviewer #2: No
---

## [Decision Letter · Decision Letter 1]

27 Sep 2023

Dear Dr. Das,

We are pleased to inform you that your manuscript 'Symptomatic and asymptomatic enteric protozoan parasitic infection and their association with subsequent growth parameters in under five children in South Asia and sub-Saharan Africa' has been provisionally accepted for publication in PLOS Neglected Tropical Diseases.

Best regards,

Luther A Bartelt

Academic Editor

Charles Jaffe

Section Editor

Thank you for addressing the reviewer's concerns. In the final version, please do clarify or remove this sentence "If the risk of child malnutrition irrespective of diarrhea symptoms differs by individual protozoan species, intervention programs aimed at improving nutritional status may have varying potential for reducing child morbidity and mortality".

Reviewer's Responses to Questions

**Key Review Criteria Required for Acceptance?**

**Methods**

-Are the objectives of the study clearly articulated with a clear testable hypothesis stated?

-Is the study design appropriate to address the stated objectives?

-Is the population clearly described and appropriate for the hypothesis being tested?

-Is the sample size sufficient to ensure adequate power to address the hypothesis being tested?

-Were correct statistical analysis used to support conclusions?

-Are there concerns about ethical or regulatory requirements being met?

Reviewer #2: Previous comments are acceptably clarified

**Results**

-Does the analysis presented match the analysis plan?

-Are the results clearly and completely presented?

-Are the figures (Tables, Images) of sufficient quality for clarity?

Reviewer #2: Previous comments are acceptably clarified

**Conclusions**

-Are the conclusions supported by the data presented?

-Are the limitations of analysis clearly described?

-Do the authors discuss how these data can be helpful to advance our understanding of the topic under study?

-Is public health relevance addressed?

Reviewer #2: Previous comments are acceptably clarified

**Editorial and Data Presentation Modifications?**

Reviewer #2: Authors have responded well to my previous comments. However, I remain puzzled about their sentence "If the risk of child malnutrition irrespective of diarrhea symptoms differs by individual protozoan species, intervention programs aimed at improving nutritional status may have varying potential for reducing child morbidity and mortality". This is is still quite cryptic and provide little extra in the context of the sentence before it.

I would advice the authors to be a bit more concrete, or else delete this sentence. If they suggest that specific measures may be beneficial that are dependent on the outcome of their study, such as limiting protozoan transmission or providing specific protozoan treatment, I think they could say so.

I also think that authors should make it very clear in the abstract that the growth parameters reported is based on a 60 day follow-up after diagnosing the infection. For example by adding in line 37 " ...child anthropometric outcomes _after two months_ among children..."

**Summary and General Comments**

Reviewer #2: This is a valuable and thorough study of an available large, high quality dataset.

PLOS authors have the option to publish the peer review history of their article (what does this mean?). If published, this will include your full peer review and any attached files.

Reviewer #2: **Yes: **Kurt Hanevik

---

## [Editor Report · Acceptance letter]

5 Oct 2023

Dear Dr. Das,

We are delighted to inform you that your manuscript, " Symptomatic and asymptomatic enteric protozoan parasitic infection and their association with subsequent growth parameters in under five children in South Asia and sub-Saharan Africa," has been formally accepted for publication in PLOS Neglected Tropical Diseases.

Best regards,

Shaden Kamhawi

co-Editor-in-Chief

Paul Brindley

co-Editor-in-Chief
